# Critical Assessment of the Prospects of Quorum-Quenching Therapy for *Staphylococcus aureus* Infection

**DOI:** 10.3390/ijms24044025

**Published:** 2023-02-16

**Authors:** Michael Otto

**Affiliations:** Pathogen Molecular Genetics Section, Laboratory of Bacteriology, National Institute of Allergy and Infectious Diseases, National Institutes of Health, Bethesda, MD 20814, USA; motto@niaid.nih.gov; Tel.: +1-301-761-6402

**Keywords:** *Staphylococcus aureus*, anti-virulence, quorum-sensing, quorum-quenching, Agr, animal infection models

## Abstract

*Staphylococcus aureus* is an important pathogen that causes a high number of infections and is one of the leading causes of death in hospitalized patients. Widespread antibiotic resistance such as in methicillin-resistant *S. aureus* (MRSA) has prompted research into potential anti-virulence-targeted approaches. Targeting the *S. aureus* accessory gene regulator (Agr) quorum-sensing system, a master regulator of virulence, is the most frequently proposed anti-virulence strategy for *S. aureus*. While much effort has been put into the discovery and screening for Agr inhibitory compounds, in vivo analysis of their efficacy in animal infection models is still rare and reveals various shortcomings and problems. These include (i) an almost exclusive focus on topical skin infection models, (ii) technical problems that leave doubt as to whether observed in vivo effects are due to quorum-quenching, and (iii) the discovery of counterproductive biofilm-increasing effects. Furthermore, potentially because of the latter, invasive *S. aureus* infection is associated with Agr dysfunctionality. Altogether, the potential of Agr inhibitory drugs is nowadays seen with low enthusiasm given the failure to provide sufficient in vivo evidence for their potential after more than two decades since the initiation of such efforts. However, current Agr inhibition-based probiotic approaches may lead to a new application of Agr inhibition strategies in preventing *S. aureus* infections by targeting colonization or for otherwise difficult-to-treat skin infections such as atopic dermatitis.

## 1. Introduction

*Staphylococcus aureus* is a leading global pathogen that causes a considerable and hard-to-estimate number of moderately severe skin infections, but also more severe and sometimes fatal infections of the blood, bones, and lungs [1]. Most of the latter occur in the hospital, and despite a recent drop in *S. aureus* hospital-associated infections, *S. aureus* remains one of the biggest threats to immune-compromised patients, those undergoing surgery, or those with any kind of indwelling medical device. In the U.S., the fatality rate due to *S. aureus* sepsis alone has been at ~20,000 deaths annually in recent years [2]. Furthermore, widespread resistance to some of the best available anti-staphylococcal agents, such as penicillins and methicillin (methicillin-resistant *S. aureus*, MRSA), significantly increase the mortality, morbidity, and costs due to *S. aureus* infections [3].

With antimicrobial resistance on a steady rise, there is great interest in finding alternatives to antibiotics. These alternatives include, for example, vaccines, phage therapy, and anti-virulence approaches. For Gram-positive bacteria such as *S. aureus*, the situation may not be as dire as for some recently developed pan-resistant Gram-negative bacteria. Nevertheless, any non-antibiotic-based drug with therapeutic potential by itself or in combination with anti-staphylococcal antibiotics would be of great clinical use to combat *S. aureus* infections. This is especially true for those caused by MRSA, because the antibiotics to which MRSA remains susceptible are by far not as efficient as methicillin. Phage therapy is controversial for many reasons not to be discussed here; and despite great efforts over many years, there is no working vaccine for *S. aureus* [4,5]. Anti-virulence approaches have therefore been the focus of pre-clinical research aiming to find alternatives to antibiotics to treat *S. aureus* infections [6].

Many efforts have been taken to counter the effect of alpha-toxin and the many leukotoxins of *S. aureus* with monoclonal antibodies [7], but results are still outstanding or trials have been dropped probably due to early sobering results. Other anti-virulence strategies against *S. aureus* are all still in the pre-clinical stage. They comprise specific drugs against selected *S. aureus* virulence factors, such as the *S. aureus* pigment, staphyloxanthine [6,8]. Due to the fact that virtually all toxins and many other virulence factors of *S. aureus* are controlled by the quorum-sensing system Agr (which stands for accessory gene regulator) [9,10], much effort has been put into so-called quorum-quenching approaches, i.e., strategies to inhibit quorum-sensing control of *S. aureus* [11,12,13]. Of note, there is no other established quorum-sensing system of *S. aureus* than Agr. Claims of alleged quorum-sensing control by the so-called “RNAIII-inhibiting peptide” (RIP)/“Target of RAP”(TRAP) system was not supported by further studies [14,15] and whether the LuxS system, often claimed to be a “universal” quorum-sensing system, has any but a metabolic role in *S. aureus* remains uncertain [16].

Much original research and many reviews have been published on Agr, its mechanism, and its regulatory roles [17,18,19]. There have also been many reviews on pre-clinical research aimed to target Agr [6,11,12,13,15,20]. However, these have for the most part emphasized the potential of the approach based on the knowledge of Agr’s regulatory roles, the theoretical drug targets in the Agr system, and the in vitro analysis of Agr quorum-quenching activity by specific drugs. The assessment of in vivo efficacy of such Agr-inhibitory substances has barely reached beyond a level of “proof-of-principle” assessment. However, with Agr interference having first been reported in 1997 [21], there now have been almost three decades during which there have been considerable efforts to evaluate Agr-inhibitory drugs in animal infection models. We are thus at a point at which we should not solely aim to assess the in vivo potential of Agr-inhibitory drugs in terms of demonstration of “proof-of-principle”, or alleged promise that is based predominantly only on the investigation in skin infection models [22] but subject those studies to a more critical assessment.

In this review, I will briefly outline the Agr system and its “druggability”. Based on a comprehensive review of the available literature, I will then present studies that attempted to demonstrate the efficacy of Agr inhibitors in animal infection models and evaluate them. Finally, I will conclude with an outlook on where we stand regarding *S. aureus* quorum-quenching drugs and what remains to be done to potentially make them clinically valuable.

## 2. Quorum Sensing in *Staphylococcus aureus*: The Agr System

The Agr system is undoubtedly the best-studied staphylococcal regulatory system, both in terms of its quorum-sensing mechanism and regarding its regulon and mechanisms of target gene control [18,19,23]. It consists of an operon of four genes, *agrB*, *agrD*, *agrC*, and *agrA*, which form the quorum-sensing circuit (Figure 1). They are encoded adjacent to a regulatory RNA that is transcribed in the opposite direction and which is responsible for the control of most of the Agr regulon [18]. AgrB is the enzyme that post-translationally modifies the gene product of the *agrD* gene to form a thiolactone ring. Four different subgroups of Agr exist with considerable differences in the amino acid sequence of, and to some extent, length of the mature autoinducing peptide (AIP) [21]. The AIP can be a hepta- to nonapeptide which is exported likely also by AgrB and trimmed at the N-terminus with the help of the non-Agr-encoded membrane-located protease MroQ, at least for the AIPs of groups 1, 2, and likely 4, while the proteolytic maturation of the product of the AgrB thiolactone-introducing step remains unknown for subgroup 3 [24,25]. The AIP activates the AgrC-AgrA two-component system by binding to the membrane histidine kinase AgrC, which in turn phosphorylates AgrA, a DNA-binding response regulator that when phosphorylated binds to and activates the promoters driving transcription of *agrBDCA* and RNAIII [18]. AgrA also directly binds to the promoters of phenol-soluble modulin (PSM) genes, which are under exceptionally direct Agr regulation in contrast to other Agr-regulated genes that are controlled via RNAIII [26]. The characteristic phenotype of quorum-sensing is established by an initially low activity of the *agrBDCA* locus, which is strongly enhanced by the quorum-sensing feedback loop upon the accumulation of the AIP in a densely grown bacterial culture. The main biological purpose of this control in *S. aureus* is assumed to consist in withholding the production of toxins and secreted aggressive enzymes until the bacterial infection has grown to a stage when nutrients become scarce and tissue degradation is needed, and when the bacteria can withstand the inflammatory defensive mechanisms of the host that are triggered by these factors [18,19]. Interestingly, recent research also has established an essential role for Agr in the colonization of the skin and the intestine, for which the biological purpose remains to be investigated [27,28].

An interesting feature of the Agr system is that the different subgroups produce AIPs that inhibit the Agr circuit of other subgroups, except in rare cases when the AIPs are very similar (subgroups 1 and 4) [21]. Similar cross-inhibiting activity is generally observed for AIPs from other staphylococcal species, such as *S. epidermidis* [29,30].

## 3. Impact of Agr Control on *S. aureus* Infection and Colonization

The significant contribution of Agr to the progression of *S. aureus* infection has been demonstrated in several animal models, including infective endocarditis [31], skin and soft tissue infections including atopic dermatitis [9,28,32], pneumonia [33,34,35], and septic arthritis [36] and osteomyelitis [37]. These results are in good accordance with Agr control of major players that have been shown to drive these infections such as alpha-toxin, leukotoxins, and PSMs [38,39,40].

Contrastingly, Agr has the opposite effect on infections that are chronic and involve biofilms, such as infections of indwelling medical devices, prosthetic joint infections, or cystic fibrosis [19,41,42,43]. It also, somewhat paradoxically, promotes persistence in osteomyelitis [44]. This is often due to its exceptionally strict control of PSMs, which structure biofilms and lead to biofilm dispersal [45], and facilitate escape from host cells [46]. In the absence of PSMs in Agr dysfunctional mutants, biofilms grow thicker and more compact, which leads to increased resistance to leukocyte attacks and antibiotics [42,43]. In many host cells, the absence of Agr-controlled factors, including PSMs, leads to intercellular persistence [47,48]. The natural occurrence of Agr dysfunctional mutants has been observed for a long time and recently, their rise could be directly linked to the persistence of biofilm infections on indwelling devices [43].

Likely as a result of biofilm involvement in many cases of *S. aureus* bacteremia, which often originate from infections of indwelling devices, Agr dysfunctional strains are frequently isolated from *S. aureus* bacteremia [49] and Agr dysfunctionality has been shown to be associated with the unfavorable outcome of invasive *S. aureus* infection [50]. In rabbit models of indwelling medical device-associated versus non-device-associated infection, isogenic mutants in *agr* were less virulent in the former, but more virulent in the latter, in agreement with the human clinical results [51]. It was also interesting that in that study, Agr had a stronger impact on virulence in the non-device-associated model in mice than in rabbits. It was proposed that this may be due to the relatively increased impact of PSMs on infection outcome in mice, as they are not sensitive to many of the *S. aureus* leukotoxins, and to the fact that due to direct control by AgrA, Agr control of PSMs is much stronger than that of other *S. aureus* toxins [51].

Thus, while there has been much initial euphoria about targeting Agr for drug development, clinical findings and animal experiments have shown more recently that such use may be limited to specific types of infection. In particular, the potential exacerbation of chronic infection as well as device-associated bacteremia and potentially other invasive infections due to the impact on biofilms represent an important caveat. Furthermore, the impact of Agr on acute virulence may be exaggerated in some mouse models.

More recently, Agr has also been implicated in non-symptomatic colonization by *S. aureus* [27,28]. From a therapeutic point of view, this is of importance as only about one- third of the human population is colonized by *S. aureus*, and colonization is associated with susceptibility to infection (as shown for nasal and intestinal *S. aureus* colonization) [52,53]. While no data appear to be available for the nose, potentially indicating that efforts to demonstrate a role of Agr in nasal colonization in animal models have failed, recent research indicates a role of Agr in skin and intestinal colonization [27,28] that may be employed for the purpose of decolonization and prevention of infection.

## 4. Main Targets in the Agr System: AgrA and AgrC

Targeting the Agr quorum-sensing control has so far focused on two targets in the Agr regulatory circuit: the membrane-located histidine kinase AgrC and its cognate cytoplasmic response regulator, AgrA. Notably, interrupting the Agr circuit at any point leads to efficient inhibition of the Agr system due to its reliance on feedback activation.

Inhibition of AgrC aims to decrease the activation by its cognate AIP. Most drugs that inhibit AgrC are structural analogs of the AIP, many of them derived from the cross-inhibiting AIPs produced by *S. aureus* and other staphylococci [21,29]. These potential drugs thus mostly stem from rational design. One of the most important problems in designing AIP analogs as AgrC inhibitors lies in overcoming the labile thiolactone bond while still maintaining inhibitory activity [54,55]. However, some AgrC inhibitors were also found by screens or coincidentally, such as solonamide B or fengycins, respectively. These compounds share some structural features with AIPs, most notably a cyclic peptide part [27,56].

Drugs targeting the other main drug target in the Agr system, AgrA, are mostly derived from screens, many of which used whole-cell Agr promoter screens or in silico docking with AgrA structural data. The AgrA inhibitors found with these screens are structurally unrelated. They are in general quite hydrophobic, which is not surprising as they must pass through the cytoplasmic membrane, in contrast to inhibitors of AgrC, which attack from the outside and are generally more hydrophilic. An important consequence of the hydrophobicity is a high probability of effects that are unrelated to Agr inhibition. Many of these compounds show bactericidal effects at concentrations frequently only marginally higher than those needed for efficient inhibition of Agr, and a distinction between true quorum-quenching versus generally bactericidal or other off-target effects is often problematic. Optimally, Agr specificity is assessed by genome-wide transcriptional profiling. Except for in rare cases [57,58], this has almost never been performed. In a study on apicidin, considerable deviation from a mere Agr effect was detected, suggesting strong off-target effects [58].

Other components of the Agr system that have been targeted by drug development approaches are AIP biosynthesis via AgrB as well as AIP activity directly via anti-AIP antibodies [59]. Ambuic acid, the drug reported to inhibit AIP biosynthesis was originally found among fungal extracts when screening for activity against Fsr, the Agr homologue of *Enterococcus faecalis* [60].

## 5. Criteria Defining a Therapeutically Promising Quorum Quenching Substance

Anti-virulence pre-clinical drug development has benefited from the perceived novelty and “elegance” of the approach, often leading to the neglect of basic tests, such as toxicity and basic pharmacokinetics, that are otherwise common for antibacterial drugs. Of note, efficacy should be independent of antibacterial effects, as the absence of such effects is at the very core of the claimed advantage of anti-virulence compounds, for example regarding resistance development.

While a thorough exclusion of off-target effects such as by whole-genome transcriptional profiling is optimal, at least a rigorous test to exclude any growth effects of the quorum-quenching drug is indispensable. Among the studies that I will discuss in detail in the following, several fail to fulfill this requirement already at a very early, in vitro stage. Some studies knowingly use drugs with considerable antibacterial effects and do not even try to distinguish them from quorum-quenching effects. Interestingly, some among those even claim antibiofilm effects, which directly contradicts what we know about the relationship of Agr with biofilms, indicating strong off-target effects. Other studies recognize the need for growth controls, but only measure end optical density values when cultures are in the stationary growth phase for a long time. Such readouts cannot adequately test for growth effects in earlier growth stages, which can be considerable. To be considered adequate in ruling out the antibacterial effects of the quorum-quenching drug, whole growth curves are indispensable.

However, even among the more high-level studies that have performed a correct assessment of the absence of antibacterial effects in such a fashion, there are frequently substantial problems when determining the drug’s efficacy in animal infection models. For example, one model to assess quorum-quenching drug in vivo efficacy that has been widely used in the *S. aureus* field is a dermonecrosis model, in which the drug is pre-mixed with the bacterial inoculum before inoculation [32,61]. While premixing avoids the technically more challenging separate application of the drug in this topical infection model, this procedure is problematic for two reasons: First, the application of a drug at the time of infection does not reflect a clinical scenario. This is especially crucial in the *S. aureus* dermonecrosis model because the disease phenotypes develop very early and then slowly decline. For a prophylactic application, the drug should be given before, or for a therapeutic application, after the inoculum. Second, premixing leads to exposure of the bacteria to the drug for an undefined time in a non-in vivo environment. Particularly problematic is that in virtually all published cases, the concentration in this premix was either antibacterial as per the in vitro growth tests reported in the respective study or exceeded the concentrations that had been tested for antibacterial effects. Consequently, it is not clear whether in all the studies that used this model the achieved in vivo effects were due to quorum-quenching as opposed to mere antibacterial effects.

These limitations are equally important for systemic infection models. However, in systemic models, serum concentrations may be assessed to determine that the Agr inhibitor does not reach antibacterial concentrations in vivo—even if the drug is injected in higher concentrations. While generally missing from studies using systemic infection models to test Agr inhibitors, this has been done in other anti-virulence *S. aureus* infection studies for example in a study by Gao et al. examining in vivo effects of a ClpP-inhibiting drug [62]. Finally, modern imaging techniques combined with fluorescent or luminescent reporter constructs may be used to ascertain that Agr inhibition occurs in vivo, which some studies have performed, for example, several of those from the Horswill lab [58,63].

## 6. Assessment of Studies Investigating In Vivo Efficacy of Agr Inhibitory Drugs

To find and evaluate studies on *S. aureus* quorum-quenching efforts with in vivo efficacy assessment, PubMed was searched using the terms “*Staphylococcus aureus*” AND “Agr” and “infection” as well as “*Staphylococcus aureus*” AND “quorum sensing” AND “inhibitor” on 4 October 2022, which returned over 1400 publications. These were screened for those that used animal models to evaluate the efficacy of defined drugs with claimed Agr quorum-quenching activity. Not included were reviews, studies that used undefined extracts, studies that focused on toxins or other virulence factors with mere speculation on Agr involvement, vaccine-based efforts, and approaches using whole bacteria (probiotic approaches). Additionally, not included were publications using the ill-defined peptide RIP and derivatives, owing to the situation explained above. One publication that fit the criteria but was not included in the search results was also included.

### 6.1. General Remarks

The studies were evaluated focusing on the most commonly detected problems. First, occasionally studies failed to appropriately determine Agr inhibition by the substance in question in vitro. Only tests that measured the activity of the Agr system directly, such as by activity test of the Agr P3 promoter or by qRT-PCR of *agr* transcripts, were deemed appropriate. Indirect assessment, such as by measurement of the production of Agr-regulated proteins, was not deemed appropriate due to lack of specificity. Second, the exclusion of growth effects in the animal experiments was judged appropriate if (i) in vitro growth tests were performed using entire growth curves and no growth defect was shown over the entire growth curve, and (ii) bacteria did not come into contact with concentrations of the drug that exceeded those for which absence of growth effects was shown, before or after application. To that end, drug concentrations in the frequently used premixes or in the mouse in case of systemic application (assuming a mouse body volume of 20 mL) were calculated. Previous publications were considered if the publication showing the animal models did not include growth tests. Third, only injection of the drug before or after inoculation with bacteria was deemed appropriate. Finally, the respective tables list whether the specificity of Agr inhibition was determined to be specific by any means, such as by measuring transcription of selected other genes, or by whole transcriptome analyses, and whether the toxicity of the drug was assessed. As for the latter, any sort of cellular or in vivo assessment was deemed acceptable, notwithstanding that for a thorough pre-clinical analysis of toxicity, these assessments in virtually all cases would have to be expanded.

It is important to note that most studies only used different skin infection or wound models, while only very selected studies used models of systemic infection or infections of interior organs. With the dermonecrosis model that includes the premixing of the drug and bacteria representing the most frequently used model, a large number of studies had to be deemed not appropriate in terms of drug application. In most of them, the fact that the drug concentration in the premix—often substantially—exceeded the concentration for which the absence of growth effects had been demonstrated, represents a considerable additional problem.

### 6.2. AgrC as Target

The oldest approach to target Agr is to interfere with AIP recognition at AgrC and stems from the discovery of cross-inhibiting AIPs in 1997 [21]. Soon afterward, a cross-inhibiting AIP was used to block abscess formation by *S. aureus* [54]. In that study, the inhibitory AIP-2 was co-injected with *S. aureus* of Agr subgroup 1 (strain RN6390) and caused a significant reduction in lesion size. The somewhat paradoxical finding that AIPs significantly block abscess formation despite the lability of the thiolactone structure was rationalized in a further study using the same approach, arguing that despite the short measured in vivo lifetime of the AIP (~3 h), it can block abscess formation owing to a short initial period of Agr-dependent events in vivo [32]. Notably, there are no reports on the efficient use of the cross-inhibitory AIP approach to control any such uncomplicated skin infections. While there have not been any reports using AIP-based inhibition to control *S. aureus* infection for almost 20 years afterward, starting in 2017, there was increased interest in this approach with studies being published using cross-inhibitory AIPs, this time from other staphylococcal species [63]. In the early studies that mainly aimed to provide “proof-of-principle”, the co-injection of the bacteria with the inhibitory drug might have been excusable. However, in more recent studies, this technically problematic procedure was not changed. The failure to determine the absence of growth-inhibiting activity in these studies is also unfortunate, but it may not be as critical as for the AgrA inhibitors discussed below, because AIPs are hydrophilic and have never been reported to have any growth-inhibitory effects.

There are also two studies that reported AgrC inhibition by non-AIP-derived substances. Murray et al. used derivatives of Gram-negative quorum-sensing signals and reported that 3-tetradecanoyltetronic acid led to a decrease in pathogenesis in a mouse arthritis model [64]. However, the applied concentration was in the range of the determined MIC, so underlying growth effects are likely, and a more thorough analysis of growth effects was not performed. Baldry et al. reported that solonamide B reduced skin inflammation in a mouse model of inflammatory skin disease [65]. This study was prompted by the finding that the Agr-controlled delta-toxin drives symptoms of atopic dermatitis [66]. While these authors also used co-application of the drug with the bacteria (on the patch covering the area of inflammation), in that model this procedure is more appropriate, especially as the concentration used was much lower than the concentrations previously tested for the absence of growth effects [67]. Studies on Agr inhibitors targeting AgrC are listed in Table 1.

### 6.3. AgrA as Target

In contrast to drugs blocking AIP interaction with AgrC, those targeting AgrA must penetrate through the cytoplasmic membrane into the cytoplasm, where they interact with AgrA. This requires pronounced hydrophobicity or transporter-facilitated import. As directly demonstrated in some cases, they are then thought to inhibit the binding of AgrA to target promoters. The earliest and, according to this data, one of the most complete studies proposing an AgrA inhibitor for quorum-quenching *S. aureus* therapy is that by Sully et al. on the inhibitor savirin [*Staphylococcus aureus* virulence inhibitor, (3-(4-propan-2-ylphenyl) sulfonyl-1H-triazolo [1,5-a] quinazolin-5-one)], which was identified using an Agr P3 promoter reporter screen with a diverse library containing more than 24,000 compounds [57]. In that study, several controls were performed that are lacking in most of the more recent studies. For example, growth curves were taken comparing to both wild-type and an isogenic *agr* mutant, efficacy was demonstrated against all Agr subgroups, and whole-genome transcriptome analysis was performed showing high similarity to the changes observed comparing the wild-type and *agr* mutant indicative of the absence of off-target effects, absence of membrane-damaging activity was assessed, absence of resistance development was determined, etc. However, this study also analyzed in vivo efficacy only in a dermonecrosis/abscess model. Interestingly, the authors compared drug application by premixing—in which the drug concentration exceeded the concentration shown to be growth-neutral, as in many other studies—with an appropriately delayed application. With the delayed application, there were still significant differences in colony-forming units (CFU) and ulcer formation, but they were much less pronounced.

Savirin was also tested in a very recent study of prosthetic joint infection [70]. This study claims that savirin has a biofilm-inhibiting effect in vitro as well as on biofilm genes such as the *ica* genes coding for the polysaccharide intercellular adhesin (PIA, also called poly-N-glucosamine, PNAG). However, it is well established that the impact of Agr on biofilm formation is negative and that it has no effect on *ica* genes. Furthermore, the savirin concentrations used in vitro as well as in the in vivo application placed right after surgery considerably exceeded those for which the absence of growth effects was shown. The results in that study are thus very likely at least in part due to off-target or growth effects.

Another recent study followed up on savirin quorum-quenching by using a derivative termed staquorsin [4-Methoxy-N’-(phthalazin-1-yl) benzenesulfonohydrazide hydrochloride], which has increased Agr inhibitory and decreased growth effects as compared to savirin [71]. Unfortunately, this study is ambiguous in how the animal dermonecrosis model was performed, mentioning premixing and/or delayed application in the methods part. Like in many other studies, the concentration in the used drug/bacteria premix exceeded that for which the absence of growth effects was shown, albeit only slightly.

In a later study on another proposed AgrA-inhibitory drug called ω-hydroxyemodin by the same group that initially published on savirin, unfortunately, delayed drug application was not used anymore [72]. The same is true for a more recent, otherwise quite complete study on the drug apicidin by the Horswill group [58]. In another recent study, the diuretic drug bumetanide was shown by in silico docking and Agr promoter reporter tests to be Agr-inhibitory likely by interacting with AgrA [73]. While this study used a dermonecrosis model with appropriately delayed drug application, the study completely lacks any assessment of growth effects.

The Shoham group published extensively on biarylketone-based AgrA inhibitors that they synthesized [74] and for two of which (“F12”, “F19”) they reported efficacy in mouse wound healing, insect larva, and mouse sepsis infection models [75,76]. Unfortunately, the method by which growth was assessed was not well described and only “percentage” values were given without further explanation. These appear to indicate that growth effects may well have existed for those compounds [74]. The topical application was described to use 20 mg/kg, which is an inappropriate unit for a topical application. In the sepsis model, the drug was applied at 30 mg/kg. These concentrations considerably exceed the concentration at which already some growth effects were detected in vitro (10 mg/mL) [74].

In a very recent study, the drug hispidulin, identified by a hemolysis screen of 60 natural compounds and reported to bind to AgrA and block Agr activity, was shown to reduce mortality and bacterial load in a mouse model of pneumonia [77]. While otherwise quite complete, there is no assessment of the significance of the survival data in that study. Furthermore, the drug concentration given to the mice somewhat exceeds that for which growth effects were evaluated.

Finally, in a 2017 study, a somewhat different approach was taken by developing peptide-conjugated locked nucleic acids (PLNAs) targeting *agrA* mRNA, and efficacy was tested in a dermonecrosis model [78]. The PLNAs were pre-mixed with the bacteria in that model in that otherwise thorough study that revealed quite considerable in vivo efficacy.

Studies on Agr inhibitors targeting AgrA are listed in Table 2.

### 6.4. Drugs with Other or Unknown Targets in the Agr System

Among the studies with unknown targets or targets other than AgrA and AgrC, I will only discuss a selected few because most have considerable fundamental problems such as the failure to clearly show Agr inhibition in vitro, failure to assess growth effects even in a basic fashion, or other data clearly indicating that results are due to off-target effects.

The only study with an identified target other than AgrA or AgrC is a study by the Cech group on the drug ambuic acid [59]. This substance had previously been reported to inhibit the homologous Fsr system of *E. faecalis* by inhibiting FsrB, which is the AgrB homologue [60]. Similar results were obtained in *S. aureus*, although the exact mechanism of inhibition of AIP biosynthesis remains elusive. This study also used premixing of drug and bacteria at a concentration that exceeds that for which the absence of growth effects had been shown [79].

Finally, the Quave group has long evaluated natural plant extracts for quorum-quenching activity and identified two specific quorum-quenching triterpenoid acids. The mechanism of action of these substances remains unknown. These reports are quite thorough, but they also use the dermonecrosis model with drug pre-mixing at concentrations exceeding those evaluated for growth inhibition [80,81]. Studies on Agr inhibitors targeting AgrB or with unknown targets are listed in Table 3.

## 7. Probiotic/Bacterial Interaction and Antibody-Based Approaches

While antibody/vaccine-based approaches were excluded from the focus of this review, it is worth mentioning that the Janda group has provided some proof-of-principle evidence that monoclonal antibodies against AIP (AP4-24H11) may dampen the Agr response [88]. Unfortunately, this was only shown for AIP and target strains of Agr subgroup 4, and there has not been any follow-up with other AIPs or strains ever since. Furthermore, the animal model that was used was also only a dermonecrosis model with premixing. While no other subgroups were tested with this approach, virus-like particles (VLPs) that bound with high specificity to AP4-24H11 were used for immunization and led to a significant reduction of the abscess but no significant reduction in ulcer (lesion) size [89]. Later, the same group developed VLPs against a modified AIP-1 amino acid sequence, which showed good efficacy in reducing both abscess and ulcer sizes [90]. Other infection types were not tested.

Another interesting approach to prevent *S. aureus* infection by quorum-quenching is by the use of other bacteria to inhibit colonization in what could be called a “probiotic” approach. Despite some caution in the staphylococcal field to use “non-infectious” strains to control infectious strains, which is based on the problems surrounding the early use of *S. aureus* strain 502A for that purpose [91], recent studies on Agr interference to control colonization of *S. aureus* by coagulase-negative staphylococci may be useful in that regard, especially for the control of skin infections such as atopic dermatitis [63,92]. Furthermore, *Bacillus* species have been found to produce Agr quorum-quenching agents that when orally administered block *S. aureus* colonization in the intestine in mice [27], and as recently shown in a human trial efficiently reduce overall *S. aureus* colonization in humans [93].

## 8. Conclusions

As an *S. aureus* researcher who has performed many studies on Agr, I went from an initially enthusiastic to a recently much more cautious view of the therapeutic promise of Agr inhibitors. This is due to mainly the following issues: First, we now know that the applicability of such drugs would be very limited, given the “biofilm problem”, a problem that not only is considerable for device infection but also extends to blood infection and possibly even more infection types. Second, there are severe shortcomings in much of the research that has been performed on Agr inhibitors in terms of providing clear evidence for quorum-quenching-dependent as opposed to growth-dependent or off-target effects observed in animal infection models, and even though the first attempts to use Agr inhibitors to control infection were already taken about a quarter of a century ago. Third, the animal models that were used to test Agr inhibitors were almost exclusively models of moderately severe skin infections, for which we do not need alternative therapeutics, as such infections are commonly treated without antibiotics. In contrast, alternatives to antibiotics such as Agr inhibitors would be needed for severe invasive *S. aureus* infections such as sepsis or pneumonia, but Agr inhibitors have only been tested very rarely in such infection models; and when they were, there mostly were substantial shortcomings in the setups that leave considerable doubt as for the quorum-quenching dependence of the reported effects. Of note, effects observed in the so far predominantly used dermonecrosis model are not predictive for other *S. aureus* infection types, as their pathogenesis and the degree to which specific virulence factors drive those infections differ considerably. Whether there is promise for targeting Agr in severe, systemic infection remains unanswered and represents the most important task to address in the future. There may be more promise in using “probiotic”-type approaches to control colonization as a prerequisite of *S. aureus* infection, but this is a very recent line of research that needs to be developed further.

## Figures and Tables

**Figure 1 ijms-24-04025-f001:**
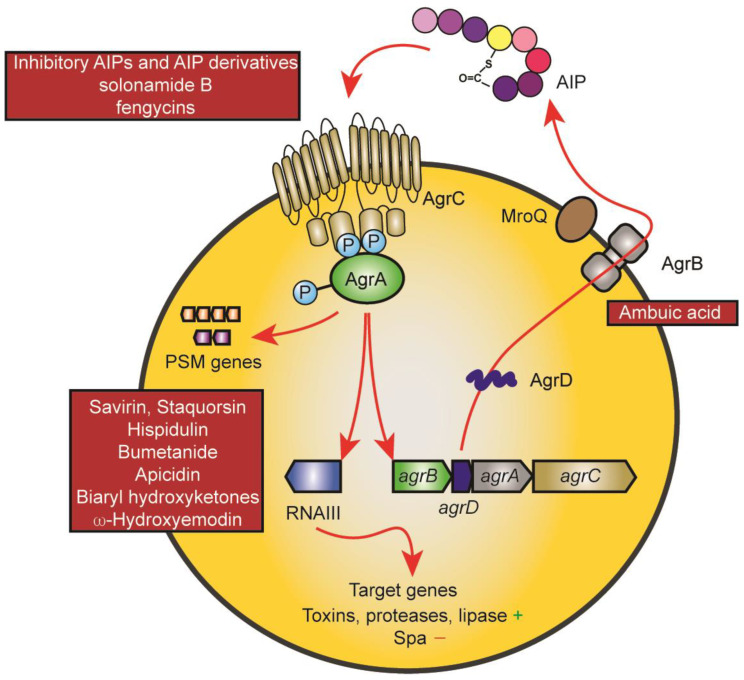
The Agr system and points of drug interference. The mechanisms underlying the Agr quorum-sensing circuit and target gene control are shown using red arrows. The AgrD AIP precursor is modified and exported by AgrB and further proteolytically trimmed by MroQ. Upon reaching a threshold concentration, the AIP leads to auto-phosphorylation of the membrane AIP sensor histidine kinase AgrC, which then leads to phosphorylation of the response regulator protein AgrA. AgrA promotes transcription from the promoters of the Agr system (*agrBDCA* promoter) and of the promoter controlling transcription of RNAIII, a regulatory small RNA that controls Agr target gene expression. PSM genes are under exceptional, direct control by AgrA promoter binding. Main points of attack by drugs are the AIP-AgrC interaction (top left) and AgrA and its interaction with target promoters (bottom left). The only drug targeting another mechanism is ambuic acid, which has been reported to interfere with AIP biosynthesis (AgrB).

**Table 1 ijms-24-04025-t001:** In vivo studies with drugs targeting AgrC.

Study Title (Year Published)	Substance	Agr Inhibition Shown ^1^	Infection Type	Appropriate Drug Application ^1^	Growth Effects Ruled out ^1^	Agr Specificity Tested ^1^	Toxicity Evaluation ^1^
Severn et al. (2022) [68]	*S. hominis* AIP-2	Yes	Dermonecrosis Epicutaneous skin infection	Yes	No	No	No
Brown et al. (2020) [69]	*S. simulans* AIP-1	Yes	Dermonecrosis	No	No	No	No
Baldry et al. (2018) [65]	Solonamide B	Yes	Atopic dermatitis	(Yes)	Yes	No	Yes
Paharik et al. (2017) [63]	*S. caprae* AIP	Yes	Dermonecrosis	No	No	No	No
Murray et al. (2014) [64]	3-tetradecanoyltetronic acid	Yes	Arthritis	? ^2^	No	No	No
Wright et al. (2005) [32]	*S. aureus* AIP-2	Yes	Dermonecrosis	No	No	No	No
Mayville et al. (1999) [54]	*S. aureus* AIP-2	Yes	Dermonecrosis	No	No	No	No

^1^ See remarks in the text as to which procedures were deemed appropriate. ^2^ No information given.

**Table 2 ijms-24-04025-t002:** In vivo studies with drugs targeting AgrA.

Study (Year Published)	Substance	Agr Inhibition Shown ^1^	Infection Type	Appropriate Drug Application ^1^	Growth Effects Ruled out ^1^	Agr Specificity Tested ^1^	Toxicity Evaluation ^1^
Pant et al. (2022) [70]	Savirin	Yes	Prosthetic joint infection	Yes	No	No	No
Ren et al. (2022) [77]	Hispidulin	Yes	Pneumonia	Yes	No	No	Yes
Mahdally et al. (2021) [71]	Staquorsin	Yes	Skin abscess	No (?) ^2^	Yes	No	Yes
Palaniappan et al. (2021) [73]	Bumetanide	Yes	Dermonecrosis	Yes	No	No	Yes
Parlet et al. (2019) [58]	Apicidin	Yes	Dermonecrosis	No	No	Yes (selected targets)	No
Greenberg et al. (2018) [75]	Biaryl hydroxyketones F12, F19	No	Sepsis	Yes	No	No	No
Kuo et al. (2015) [76]	Biaryl hydroxyketones F12, F19	No	Wound healing Insect larva	Yes	No	No	No
Da et al. (2016) [78]	Antisense locked nucleic acids	Yes	Dermonecrosis	No	Yes	No	No
Daly et al. (2015) [72]	ω-Hydroxyemodin	Yes	Dermonecrosis	No	No	No	Yes
Sully et al. (2014) [57]	Savirin	Yes	Air pouch Dermonecrosis	Yes ^3^	Yes ^3^	Yes	No

^1^ See remarks in the text as to which procedures were deemed appropriate. ^2^ No information given. ^3^ Premixing and delayed application were used in different experiments.

**Table 3 ijms-24-04025-t003:** In vivo studies with drugs targeting unknown or other targets in the Agr system.

Study Title (Year Published)	Substance	Agr Inhibition Shown ^1^	Infection Type	Appropriate Drug Application ^1^	Growth Effects Ruled out ^1^	Agr Specificity Tested ^1^	Toxicity Evaluation ^1^
Yuan et al. (2022) [82]	Luteolin (3′,4′,5,7-tetrahydroxyflavone)	No	Pneumonia	Yes	No	No	No
Khayat et al. (2022) [83]	Sitagliptin	Yes	Sepsis	No	No	Yes (results suggesting non-specificity)	No
Hu et al. (2022) [84]	Luteolin-loaded nanoparticles	No	Joint humeral implant infection	N/A	No	No	Yes
Zheng et al. (2022) [85]	Benzylaniline derivative	No	Sepsis	Yes	No	Yes (results suggesting non-specificity)	No
Mishra et al. (2021) [86]	N-4-Methoxyphenyl- 3-(4-methoxyphenyl)-propanamide	No	*Galleria* survival	Yes	No	No	Yes
Salam et al. (2021) [80]	Castaneroxy A (a hydroperoxy cycloartane triterpenoid)	Yes	Dermonecrosis	No	No	No	Yes
Tang et al. (2020) [81]	3-oxo-olean-12-en-28-oic acid, 3-oxotirucalla-7,24Z-dien-26-oic acid, 3α-hydroxytirucalla-7,24 Z-dien-27-oic acid	Yes	Dermonecrosis	No	No	No	Yes
Todd et al. (2017) [59]	Ambuic acid	Yes	Dermonecrosis	No	No	Yes (selected targets)	Yes
Yang et al. (2016) [87]	NO-releasing dexamethasone derivative	No	Sepsis	Yes	No	No	No

^1^ See remarks in the text as to which procedures were deemed appropriate.

## Data Availability

Not applicable.

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
