# Peer review of "Critical Assessment of the Prospects of Quorum-Quenching Therapy for Staphylococcus aureus Infection"

_ijms, 2023, doi:10.3390/ijms24044025_

Round 1

Reviewer 1 Report

This study is a review giving valuable insights into agr- inhibition strategies.

Minor comments:

*Please provide full phrases for the first time and abbreviations in repetitions. For example:  in lines 12, 61, 84, 93, 261, and 350 

*There is no figure 1.

*Please imply to figure 2 within the text.

*Provide a description for [?] in tables 1 and 2.

Best

Author Response

*Please provide full phrases for the first time and abbreviations in repetitions. For example:  in lines 12, 61, 84, 93, 261, and 350 

Reply: The manuscript was carefully checked and abbreviations are now explained at first use.

*There is no figure 1.

*Please imply to figure 2 within the text.

Reply: Apologies - there is only a Figure 1; this was typo that was corrected, and the figure is now cited in the text.

*Provide a description for [?] in tables 1 and 2.

Reply: Footnotes were added to that effect.

Reviewer 2 Report

Line 61: “debunked” is a prejudicial word.  “not supported by further studies” would be preferred.

Line 108: “get” is slang.  “become scare” should be used.

Lines 113-116: The comments are difficult to follow as written.  Please expand with more detail.

Lines 119 - 123: Agr has been found to play a dynamic role in abscess formation [PMID: 15665088], and this would be a good place to discuss the role of Agr in different stages of acute abscess formation as a contrast is made in the next paragraph about chronic infections.

Lines 125 - 131: While biofilms and Agr dysfunction are important for persistent infections, an equally or perhaps even more important Agr dysfunction is found with persistent S. aureus within host hosts.  Again PSMs play a role as well as loss of other lytic toxins.  The role of agr-negative strains in persistent host cell infection deserves at least a full paragraph of discussion.  Agr negative strains are also important in persistent infections in patients with cystic fibrosis and osteomyelitis.   Of course, these discussions are critical to this manuscript as the question of agr suppressive therapies may also be persistence-generating therapies.   Clearly, a risk which needs to be discussed in detail.

Lines 232-234: The considerations in these lines can be directly related to the need to define the dynamic nature of abscess formation as suggested for Lines 125-131.

Line 263: Tables 1 - 3 are very strong.

Line 304-305: This is speculation and should be dropped.  

Lines 346-347: Drop one of the “only’s”.

Lines 354-355: Prof. Otto is correct that agr does not directly control ica genes, the statement that “Agr does not control biofilm formation” is not accurate.  Agr has major impact on protease expression and dispersion of biofilms [see PMID: 18437240].  There are several other papers that show similar results.  This paragraph needs to be fully revised. 

Line 380: Again, the term “bizarre” is a prejudicial term, which could be replaced by the term “inappropriate”.

Lines 381-382: How did the author calculate the concentration?  From 30 mg/kg to 10 mg/ml, the weight of water does not work out: 1 kg  = 1,000 ml of water, which would be much less than 10 mg/ml.  This needs to be explained.

Line 436-440: While the science is very solid, it is unclear what a 64% reduction in nasal cfu’s means clinically.  When there are often log’s of bacteria on mucosal sites, it may not be accurate to state “reduce overall S. aureus colonization in humans” as a reduction in numbers of organisms is not the same as reduced colonization.  In clinical trials where antimicrobial agents are used to “decolonize” before surgical procedure, elimination of S. aureus is the goal.  Thus, this needs to be revised, especially to separate intestinal vs. nasal effects.  

Author Response

Line 61: “debunked” is a prejudicial word.  “not supported by further studies” would be preferred.

Reply: Absolutely, my apologies. Corrected as suggested.

Line 108: “get” is slang.  “become scare” should be used.

Reply: corrected as suggested.

Lines 113-116: The comments are difficult to follow as written.  Please expand with more detail.

Reply: This was re-written.

Lines 119 - 123: Agr has been found to play a dynamic role in abscess formation [PMID: 15665088], and this would be a good place to discuss the role of Agr in different stages of acute abscess formation as a contrast is made in the next paragraph about chronic infections.

Reply: I wanted to keep the description of the positive role of Agr in different infections here short, as this has been reviewed frequently in detail. It would be a little awkward to single out abscess formation and describe in detail here. However, the study the reviewer refers to and the results described therein in terms of dynamics of abscess formation are discussed later in the manuscript.

Lines 125 - 131: While biofilms and Agr dysfunction are important for persistent infections, an equally or perhaps even more important Agr dysfunction is found with persistent S. aureus within host hosts.  Again PSMs play a role as well as loss of other lytic toxins.  The role of agr-negative strains in persistent host cell infection deserves at least a full paragraph of discussion.  Agr negative strains are also important in persistent infections in patients with cystic fibrosis and osteomyelitis.   Of course, these discussions are critical to this manuscript as the question of agr suppressive therapies may also be persistence-generating therapies.   Clearly, a risk which needs to be discussed in detail.

Reply: I absolutely agree and have worked in this paragraph the role in cystic fibrosis and osteomyelitis (and relevant citations) as well as the issue of Agr/PSM-mediated host cell escape and intracellular persistence in their absence (again with relevant citations).

Lines 232-234: The considerations in these lines can be directly related to the need to define the dynamic nature of abscess formation as suggested for Lines 125-131.

Reply: Thanks for this excellent suggestion – this was worked in here now.

Line 263: Tables 1 - 3 are very strong.

Reply: Thanks! Quite some work went into compiling these tables…

Line 304-305: This is speculation and should be dropped.  

Reply: I agree – while this is probably the case, such negative results are not commonly published and it is not appropriate to speculate on this situation. The sentence was deleted.

Lines 346-347: Drop one of the “only’s”.

Reply: Thanks! One was deleted.

Lines 354-355: Prof. Otto is correct that agr does not directly control ica genes, the statement that “Agr does not control biofilm formation” is not accurate.  Agr has major impact on protease expression and dispersion of biofilms [see PMID: 18437240].  There are several other papers that show similar results.  This paragraph needs to be fully revised. 

Reply: Absolutely agree. This is a matter of wording only. I wrote that “Agr does not control biofilm formation….in that way”, meaning that it controls biofilm formation negatively (which is a major point in this review), not positively as the authors of that study argue. This was entirely reworded for clarity.

Line 380: Again, the term “bizarre” is a prejudicial term, which could be replaced by the term “inappropriate”.

Reply: changed as suggested.

Lines 381-382: How did the author calculate the concentration?  From 30 mg/kg to 10 mg/ml, the weight of water does not work out: 1 kg  = 1,000 ml of water, which would be much less than 10 mg/ml.  This needs to be explained.

Reply:  Apologies and thanks for spotting this. It should be ug not mg. The symbol font had been reverted to main text font. Thus, 30 mg/kg = 30 mg/l  = 30 ug/ml > 10 ug/ml

I should add that finding this information in this specific manuscript was extremely difficult – in addition to the lack of description of how values were exactly obtained.

Line 436-440: While the science is very solid, it is unclear what a 64% reduction in nasal cfu’s means clinically.  When there are often log’s of bacteria on mucosal sites, it may not be accurate to state “reduce overall S. aureus colonization in humans” as a reduction in numbers of organisms is not the same as reduced colonization.  In clinical trials where antimicrobial agents are used to “decolonize” before surgical procedure, elimination of S. aureus is the goal.  Thus, this needs to be revised, especially to separate intestinal vs. nasal effects.

Reply: This is probably not the place to discuss this study in detail, and it is only presented as an "outlook". But just for the information of the reviewer, I would disagree with the statement that reduced overall colonization is not adequately expressed by measuring total CFU in the human body. What else would be an accurate measurement?

As for the detailed discussion of clinical purpose, it is stated in that study that such purpose would be in a long-term fashion. The reasoning behind this is that by far the main reservoir of S. aureus in human is in the intestine (a so far completely neglected fact) and, as the study implies, this serves to replenish/recolonize other body sites - such as the nose (as shown in the study) or potentially the skin. It is absolutely understood that when short-term efficacy is needed - such as when aiming to decolonize patients before surgery – targeting the main reservoir in the gut will not achieve valuable effects and decolonization of the sites that probably serve as more direct sources of self-contamination – nose and skin – is needed.